# In-plane uniaxial pressure-induced out-of-plane antiferromagnetic moment and critical fluctuations in BaFe$_2$As$_2$

Panpan Liu[1,11], Mason L. Klemm[2,11], Long Tian[1], Xingye Lu[1 ✉], Yu Song[3], David W. Tam[2], Karin Schmalzl[4], J. T. Park[5], Yu Li[2], Guotai Tan[1], Yixi Su[6], Frédéric Bourdarot[7], Yang Zhao[8], Jeffery W. Lynn[8], Robert J. Birgeneau[3,9,10] & Pengcheng Dai[2 ✉]

A small in-plane external uniaxial pressure has been widely used as an effective method to acquire single domain iron pnictide BaFe$_2$As$_2$, which exhibits twin-domains without uniaxial strain below the tetragonal-to-orthorhombic structural (nematic) transition temperature $T_s$. Although it is generally assumed that such a pressure will not affect the intrinsic electronic/ magnetic properties of the system, it is known to enhance the antiferromagnetic (AF) ordering temperature $T_N$ ( $< T_s$) and create in-plane resistivity anisotropy above $T_s$. Here we use neutron polarization analysis to show that such a strain on BaFe$_2$As$_2$ also induces a static or quasi-static out-of-plane ($c$-axis) AF order and its associated critical spin fluctuations near $T_N/T_s$. Therefore, uniaxial pressure necessary to detwin single crystals of BaFe$_2$As$_2$ actually rotates the easy axis of the collinear AF order near $T_N/T_s$, and such effects due to spin-orbit coupling must be taken into account to unveil the intrinsic electronic/magnetic properties of the system.

[1] Center for Advanced Quantum Studies and Department of Physics, Beijing Normal University, 100875 Beijing, China. [2] Department of Physics and Astronomy, Rice University, Houston, TX 77005, USA. [3] Department of Physics, University of California, Berkeley, CA 94720, USA. [4] Forschungszentrum Jülich GmbH, Jülich Centre for Neutron Science at ILL, 71 avenue des Martyrs, 38000 Grenoble, France. [5] Heinz Maier-Leibnitz Zentrum (MLZ), Technische Universität München, 85748 Garching, Germany. [6] Jülich Centre for Neutron Science JCNS at MLZ, Forschungszentrum Jülich GmbH, Lichtenbergstr. 1, D-85747 Garching, Germany. [7] University Grenoble Alpes, CEA, IRIG, MEM-MDN, Grenoble, France. [8] NIST Center for Neutron Research, National Institute of Standards and Technology, Gaithersburg, MD 20899, USA. [9] Materials Sciences Division, Lawrence Berkeley National Laboratory, Berkeley, CA 94720, USA. [10] Department of Materials Science and Engineering, University of California, Berkeley, CA 94720, USA. [11] These authors contributed equally: Panpan Liu, Mason L. Klemm. ✉email: luxy@bnu.edu.cn; pdai@rice.edu

U nderstanding the intrinsic electronic, magnetic, and nematic properties of iron pnictides such as BaFe₂As₂ form the basis to unveil the microscopic origin of high-temperature superconductivity because the system is a parent compound of iron-based superconductors[1–5]. As a function of decreasing temperature, BaFe₂As₂ first exhibits a tetragonal-to-orthorhombic structural transition at $T_s$ and forms a nematic ordered phase, followed closely by a collinear antiferromagnetic (AF) order with moment along the $a$-axis of the orthorhombic lattice below the Néel temperature $T_N$ ($\leq T_s$) (Fig. 1a)[6–9]. Since single crystals of BaFe₂As₂ form twin domains in the orthorhombic state below $T_s$, an external uniaxial pressure applied along one of the in-plane Fe-Fe bond direction has been widely used as an effective method to acquire single domains of iron pnictide crystals and determine their intrinsic transport[10–15], electronic[16–18], and magnetic[19–21] properties (inset in Fig. 1b). Although uniaxial pressure necessary to detwin single crystals of BaFe₂As₂ is known to increase $T_N$ (Fig. 1b)[22–25] and create in-plane resistivity aniso-tropy above $T_s$[14], it is generally assumed that it only induces a small strain on the sample and does not significantly modify the electronic and magnetic properties of the system[10–19,21]. Recently, nuclear magnetic resonance (NMR) experiments have revealed that an in-plane uniaxial strain on BaFe₂As₂ induces an enhancement of the low-energy spin fluctuations along the $c$-axis in the para-magnetic state above $T_N$[26]. However, it is unclear whether the applied uniaxial pressure can actually modify the collinear AF structure of the system (Fig. 1a)[6,7].

In this work, we use polarized neutron scattering and unpolarized neutron diffraction to demonstrate that an in-plane uni-axial pressure necessary to detwin BaFe₂As₂ also induces a $c$-axis ordered magnetic moment and changes the easy axis of the col-linear AF structure around the magnetic/nematic critical scat-tering temperature regime where the applied pressure has a large impact on the crystal structure of the system (Fig. 1c–g)[27]. In addition, we find that the applied pressure induces $c$-axis polar-ized critical spin fluctuations that diverge near $T_N/T_s$, confirming the results of NMR experiments[26]. Therefore, uniaxial pressure on BaFe₂As₂ that breaks the tetragonal lattice symmetry also induces changes in the magnetic easy axis near the critical regime of the AF/nematic phase transitions, indicating that the intrinsic electronic and magnetic properties of the system near $T_N/T_s$ are much different from naive expectations.

## Results

**Collinear magnetic order in twinned BaFe₂As₂.** Without external uniaxial pressure, BaFe₂As₂ exhibits separate weakly first-order magnetic and second-order structural phase transi-tions ($T_s > T_N$ by ~0.75 K)[7]. The spins within each FeAs layer are collinear and arranged antiferromagnetically along the $a$-axis and ferromagnetically along the $b$-axis of orthorhombic structure with lattice parameters of $a$ and $b$, respectively ($a > b$). Along the out-of-plane direction, spins are arranged antiferromagnetically within one chemical unit cell (lattice parameter $c$), but have no net magnetic moment along the $c$-axis (Fig. 1a)[6,7]. For a collinear Ising antiferromagnet with second-order (or weakly first order) magnetic phase transition, magnetic critical scattering with moments polarized along the longitudinal (parallel to the ordered moment or $a$-axis) direction should diverge at $T_N$, while spin fluctuations with moments polarized transverse to the ordered moment ($b$- and $c$-axis) direction should not diverge[28–32]. Unpolarized[33] and polarized[34] neutron scattering experiments on strain-free BaFe₂As₂ confirm this expectation. While the long-itudinal component ($M_a$) of the magnetic critical scattering, defined as low-energy spin fluctuations polarized along the $a$-axis direction, diverges at $T_N$, the transverse components $M_b$ and $M_c$

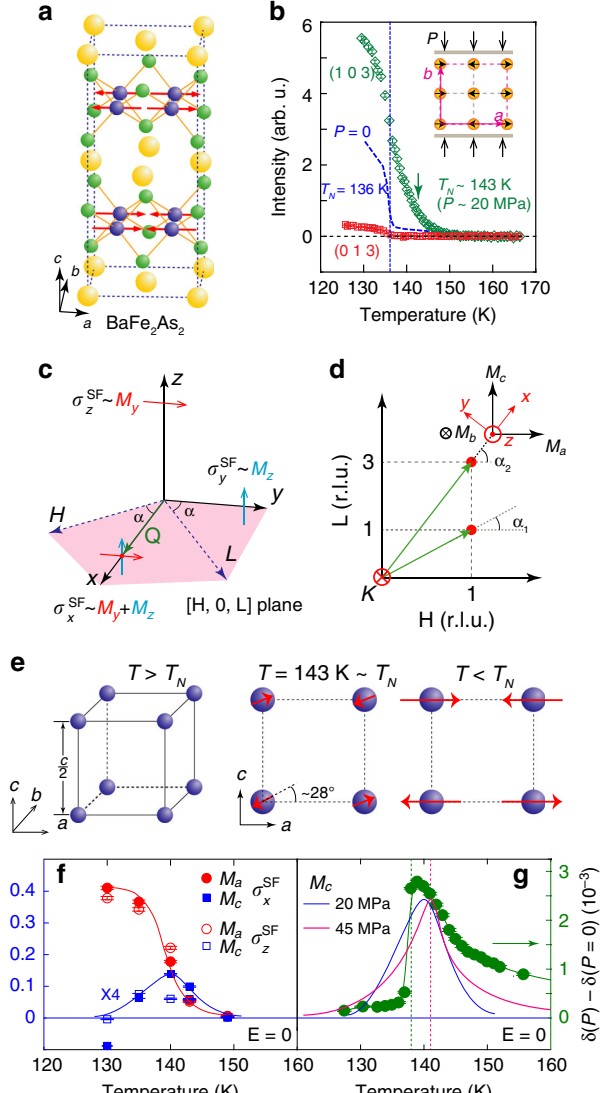

**Fig. 1 Summary of the effect of uniaxial pressure on crystalline lattice and magnetic structures of Ba₂Fe₂As₂. a** Crystal and AF structure of BaFe₂As₂. The red arrows indicate the $a$-axis direction of magnetic moments. **b** Magnetic order parameters measured at $\mathbf{Q}_1 = (1, 0, 3)$ and $\mathbf{Q} = (0, 1, 3)$ under uniaxial pressure, revealing $T_N = 143$ K. The blue dashed curve denotes the magnetic order parameter measured on a strain-free sample with $T_N = 136$ K. **c** Scattering geometry of polarized neutron scattering experiment in the $[H, 0, L]$ plane. **d** The reciprocal space, where the fluctuating moments along the $a$-, $b$-, and $c$-axis directions are marked as $M_a$, $M_b$, and $M_c$, respectively. **e** Spin arrangements of BaFe₂As₂ in the paramagnetic (left), near $T_N$ (middle), and low-temperature AF state. **f** Temperature dependence of the static ordered magnetic moment $M_a$ and $M_c$ as determined from $\sigma_{x,z}^{SF}$ at $(1, 0, 1)$ and $(1, 0, 3)$. The vertical error bars are estimated errors from fits-to-order parameters. **g** Comparison of temperature dependence of the strain-induced lattice distortion from ref. [27] and our estimated $M_c$ at ~20 (blue solid line) and ~45 (pink solid line) MPa.

along the $b$- and $c$-axis, respectively (Fig. 1c, d), do not diverge at $T_N$.

**Effect of uniaxial pressure on lattice parameters of BaFe₂As₂.** The in-plane uniaxial pressure-induced tetragonal symmetry-breaking lattice distortion $[\delta(P \neq 0) - \delta(P = 0)$, where $\delta = (a - b)/(a + b)]$ has a Curie–Weiss temperature dependence in the

paramagnetic state and peaks near $T_N/T_s$, but is greatly suppressed below $T_N/T_s$ when the intrinsic orthorhombic lattice distortion of $BaFe_2As_2$ sets in (Fig. 1g)[27]. In the paramagnetic state, NMR experiments on $BaFe_2As_2$ suggest that an in-plane uniaxial strain can induce a diverging $c$-axis polarized spin susceptibility $\chi_c''$, which equals to $M_c$ in the zero energy limit, on approaching $T_N/T_s$[26]. Since $c$-axis polarized low-energy spin fluctuations do not diverge around $T_N/T_s$ in the strain-free $BaFe_2As_2$[34], it is important to confirm the NMR results and determine if the diverging $\chi_c''$ is a precursor of a new magnetic order with a component along the $c$-axis (Fig. 1f)[28].

**Neutron polarization analysis of spin excitations in detwinned $BaFe_2As_2$.** Our polarized neutron scattering experiments were carried out on the CEA CRG-IN22 triple-axis spectrometer equipped with Cryopad capability at the Institut Laue-Langevin and the BT-7 triple-axis spectrometer at the NIST Center for Neutron Research. The experimental setup for IN22 has been described in detail before[34–39], while polarized neutrons were controlled and analyzed using a polarized $^3He$ filter on BT-7[40,41]. We have also carried out unpolarized neutron diffraction experiments on BT-7 using an in-situ uniaxial pressure device[25]. The wave vector transfer $\mathbf{Q}$ in reciprocal space in $\text{Å}^{-1}$ is defined as $\mathbf{Q} = H\mathbf{a}^* + K\mathbf{b}^* + L\mathbf{c}^*$, with $\mathbf{a}^* = (2\pi/a)\hat{\mathbf{a}}$, $\mathbf{b}^* = (2\pi/b)\hat{\mathbf{b}}$, and $\mathbf{c}^* = (2\pi/c)\hat{\mathbf{c}}$, where $a \approx b \approx 5.6\,\text{Å}$, $c = 12.96\,\text{Å}$, and $H$, $K$, $L$ are Miller indices. In this notation, the collinear AF structure of $BaFe_2As_2$ in Fig. 1a gives magnetic Bragg peaks at $(H, K, L) = (1, 0, L)$ with $L = 1, 3, \ldots$. The magnetic responses of the system at a particular $\mathbf{Q}$ along the orthorhombic lattice $a$-, $b$-, and $c$-axis directions are marked as $M_a$, $M_b$, and $M_c$, respectively (Fig. 1a–d). The scattering plane is $[H, 0, L]$. The incident neutrons are polarized along the $\mathbf{Q}$ ($x$), perpendicular to $\mathbf{Q}$ but in the scattering plane ($y$), and perpendicular to both $\mathbf{Q}$ and the scattering plane ($z$) (Fig. 1c). In this geometry, the neutron spin-flip (SF) scattering cross sections $\sigma_x^{SF}$, $\sigma_y^{SF}$, and $\sigma_z^{SF}$ are related to the components $M_a$, $M_b$, and $M_c$ via $\sigma_x^{SF} = \frac{R}{R+1}M_y + \frac{R}{R+1}M_z + B$, $\sigma_y^{SF} = \frac{1}{R+1}M_y + \frac{R}{R+1}M_z + B$, and $\sigma_z^{SF} = \frac{R}{R+1}M_y + \frac{1}{R+1}M_z + B$, where $R$ is the flipping ratio ($R = \sigma_{Bragg}^{NSF}/\sigma_{Bragg}^{SF} \approx 13$), $B$ is the background scattering, $M_y = \sin^2\alpha M_a + \cos^2\alpha M_c$ with $\alpha$ being the angle between $[H, 0, 0]$ and $\mathbf{Q}$, and $M_z = M_b$ (Fig. 1d)[34–39].

Figure 1b compares the temperature dependencies of the $(1, 0, 3)$ magnetic Bragg peak for strain-free and strained $BaFe_2As_2$. At zero external pressure ($P = 0$ and strain-free), the magnetic scattering shows an order parameter like increase below $T_N = 136\,\text{K}$[34]. When an uniaxial pressure of $P \approx 20\,\text{MPa}$ is applied along the $b$-axis of $BaFe_2As_2$, the Néel temperature of the sample increases to $T_N = 143\,\text{K}$[25]. The vanishingly small magnetic scattering intensity at $\mathbf{Q} = (0, 1, 3)$ suggests that the sample is essentially ~100% detwinned (Fig. 1b).

Figure 2a, b shows the energy dependence of $\sigma_x^{SF}$, $\sigma_y^{SF}$, and $\sigma_z^{SF}$ in the zero pressure paramagnetic state at $T \approx 1.015T_N \approx 138\,\text{K}$ for magnetic positions $(1, 0, 1)$ and $(1, 0, 3)$[34]. Figure 2c, d shows identical scans as those of Fig. 2a, b, respectively, in the paramagnetic state at $T \approx 1.014T_N \approx 145\,\text{K}$ with uniaxial pressure of $P \approx 20\,\text{MPa}$. Comparison of the Fig. 2a, c reveals that $\sigma_z^{SF}$ is clearly larger than $\sigma_y^{SF}$ below ~5 meV in the uniaxial strained sample. Since the $(1, 0, 1)$ peak corresponds to $\alpha_1 = 23.4°$ giving $M_y \approx 0.16M_a + 0.84M_c$ (Fig. 1d)[34], the increased $\sigma_z^{SF}$ in strained $BaFe_2As_2$ is mostly due to the increased $M_c$. For the $(1, 0, 3)$ peak, which corresponds to $\alpha_2 = 52.4°$, $M_y \approx 0.63M_a + 0.37M_c$, and the scattering is therefore much less sensitive to strain-induced changes in $M_c$. To conclusively determine the effect of uniaxial pressure on $M_c$, we consider $\sigma_z^{SF} - \sigma_y^{SF} \propto M_y - M_b$. Since $M_b$ (or

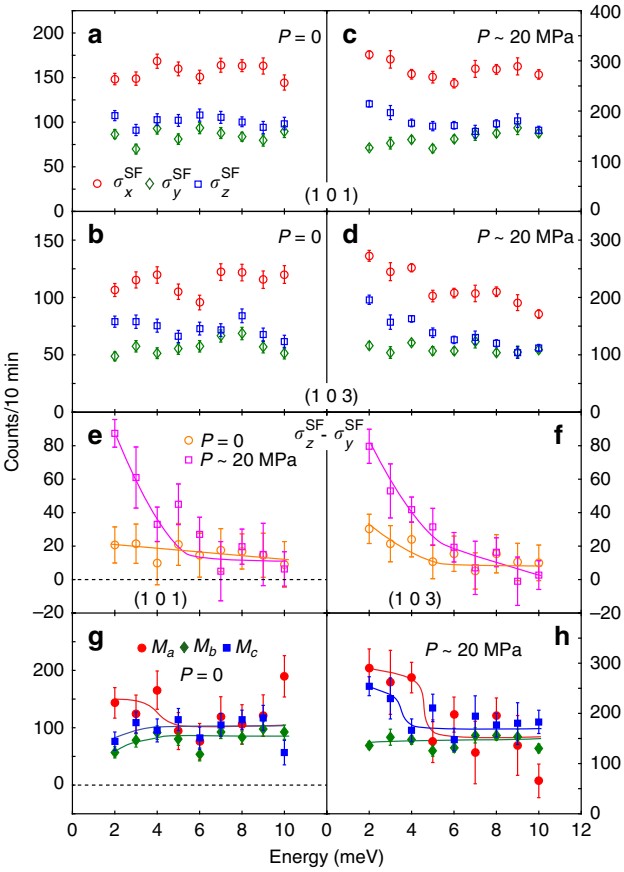

**Fig. 2 The energy dependence of the neutron spin-flip magnetic scattering near $T_N$ without and with uniaxial pressure.** Energy scans of $\sigma_x^{SF}$ (red circle), $\sigma_y^{SF}$ (green diamond), and $\sigma_z^{SF}$ (blue square) under **a, b** $P = 0$ and $T = 138\,\text{K}$[34] and **c, d** ~20 MPa and $T = 145\,\text{K}$ at the two AF wave vectors $\mathbf{Q}_1 = (1, 0, 1)$ and $\mathbf{Q}_2 = (1, 0, 3)$. **e, f** Comparison of $P = 0$ and $P \approx 20\,\text{MPa}$ ($\sigma_z^{SF} - \sigma_y^{SF}$) at $(1, 0, 1)$ and $(1, 0, 3)$. **g, h** Energy dependence of $M_a$, $M_b$, and $M_c$ extracted from the raw data in (**a–d**). The solid lines are guides to the eyes and the error bars represent 1 standard deviation.

$\sigma_y^{SF}$) does not diverge at $T_N$ or change as a function of uniaxial pressure as seen in NMR[26] and neutron polarization analysis (Fig. 2a–d), the effect of uniaxial pressure can be seen directly from the energy dependence of $\sigma_z^{SF} - \sigma_y^{SF}$ at $(1, 0, 1)$ (Fig. 2e) and $(1, 0, 3)$ (Fig. 2f). Without uniaxial pressure, $\sigma_z^{SF} - \sigma_y^{SF}$ does not diverge at the $\mathbf{Q}_1 = (1, 0, 1)$ position but diverges at $\mathbf{Q}_2 = (1, 0, 3)$ at low energies consistent with the expectation that spin fluctuations at $(1, 0, 1)$ are mostly probing $M_c$. With uniaxial pressure, we see clear divergence of low-energy spin fluctuations at $(1, 0, 1)$ below ~5 meV, thus unambiguously confirming the uniaxial pressure-induced $M_c$ enhancement around $T_N$ observed in NMR experiments[26]. To further clarify the energy dependence of $M_a$, $M_b$, and $M_c$, we estimate these components from measurements at the $(1, 0, 1)$ and $(1, 0, 3)$ positions as described in ref. [34]. By comparing the energy dependence of $M_a$, $M_b$, and $M_c$ in strain-free (Fig. 2g) and strained (Fig. 2h) $BaFe_2As_2$, we again see that the effect of uniaxial strain is to enhance $M_c$ below about 4 meV, consistent with the NMR measurements which probe $M_c$ or $\chi_c''$ in the zero energy limit[26].

To further demonstrate the effect of uniaxial strain on the magnetic critical scattering of $BaFe_2As_2$, we show in Fig. 3 the temperature dependence of $\sigma_x^{SF}$, $\sigma_y^{SF}$, and $\sigma_z^{SF}$ at $E = 2\,\text{meV}$ for the strain-free (Fig. 3a, b)[34] and strained (Fig. 3c, d) samples. At

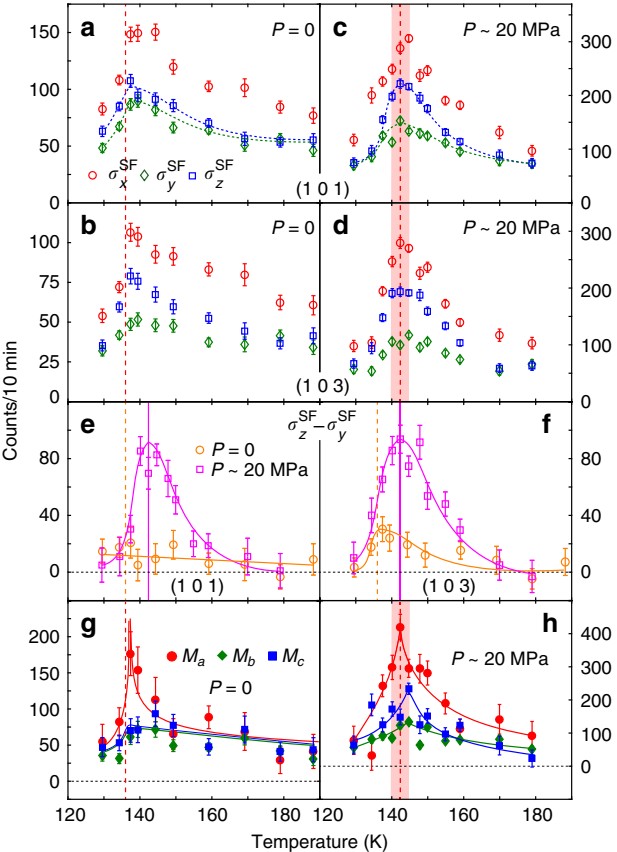

**Fig. 3 Temperature dependence of the magnetic scattering across $T_N$ at $E = 2$ meV without and with uniaxial pressure.** Temperature dependence of $\sigma_x^{SF}$, $\sigma_y^{SF}$, and $\sigma_z^{SF}$ at $E = 2$ meV of **a**, **b** uniaxial pressure-free[34] and **c**, **d** pressured ($P \approx 20$ MPa) BaFe$_2$As$_2$ at **a**, **c** (1, 0, 1) and **b**, **d** (1, 0, 3). **e**, **f** Comparison of $P = 0$ and $P \approx 20$ MPa ($\sigma_z^{SF} - \sigma_y^{SF}$) at $\mathbf{Q_1} = (1, 0, 1)$ and $\mathbf{Q_2} = (1, 0, 3)$. Temperature dependence of $M_a$, $M_b$, and $M_c$ at $E = 2$ meV for **g** uniaxial pressure-free and **h** pressured sample estimated from the data in (**a**–**d**). The dotted and solid lines are guides to the eye and the error bars represent 1 standard deviation. The vertical dashed and solid lines mark $T_N/T_s$ at $P = 0$ and $P \approx 20$ MPa, respectively.

$\mathbf{Q_1} = (1, 0, 1)$, uniaxial strain clearly enhances $\sigma_z^{SF}$ around $T_N/T_s$, where $\sigma_z^{SF} \approx M_y \approx 0.16M_a + 0.84M_c$, again consistent with the strain enhanced $\chi_c''$ in the NMR measurements[26]. Figure 3b, d shows similar measurements at $\mathbf{Q_2} = (1, 0, 3)$, which reveal much less enhancement of $\sigma_z^{SF}$ because $\sigma_z^{SF} \approx M_y \approx 0.63M_a + 0.37M_c$. Figure 3e shows temperature dependence of $\sigma_z^{SF} - \sigma_y^{SF}$ across $T_N$ at the (1, 0, 1) peak without and with uniaxial pressure. Since spin fluctuations at the (1, 0, 1) position are mostly sensitive to $M_c$, we see no divergence across $T_N$ in zero pressure case. Upon application of a ~20 MPa uniaxial pressure, the scattering clearly reveals a diverging behavior at the pressured enhanced $T_N$ (solid vertical line) (Fig. 3e). Similar measurements at the (1, 0, 3) position, which is more sensitive to $M_a$, show diverging magnetic scattering at $T_N$ with and without uniaxial pressure consistent with the NMR results (Fig. 3f)[26]. Figure 3g, h shows the temperature dependencies of the estimated $M_a$, $M_b$, and $M_c$ for strain-free and strained BaFe$_2$As$_2$, respectively, using the data in Fig. 3a–d. Comparing with the normal behavior of the strain-free BaFe$_2$As$_2$ (Fig. 3g), the $M_c$ in strained BaFe$_2$As$_2$ clearly diverges around $T_N/T_s$ (Fig. 3h), although the relative uncertainties appear more significant for $M_a$, $M_b$, and $M_c$ due to the propagation of errors (see Supplementary information for additional data and analysis).

**Effect of uniaxial pressure on static AF order of BaFe$_2$As$_2$.** In principle, a diverging dynamic spin susceptibility in the paramagnetic state of a system is an indication of the eventual magnetic order below $T_N$[28–32]. For strain-free BaFe$_2$As$_2$, the magnetic ordered moment is along the $a$-axis with no net moment along the $b$-axis and $c$-axis directions[6,7]. Therefore, only the $M_a$ component of the spin susceptibility diverges at $T_N$ (Fig. 3e–g)[34]. The observation of a diverging $M_c$ in strained BaFe$_2$As$_2$, in addition to the usual diverging $M_a$ (Fig. 3e, f, h), suggests that the applied strain may induce static magnetic ordered moment along the $c$-axis. To test this hypothesis, we carried out polarized neutron diffraction measurements on BaFe$_2$As$_2$ as a function of uniaxial pressure, focusing on the temperature and neutron polarization dependence of the scattering at $\mathbf{Q} = (1, 0, L)$ with $L = 0, 1, 2, 3,$ and 5. At wave vectors (1, 0, 0) and (1, 0, 2), there is no evidence of magnetic scattering, consistent with uniaxial pressure-free BaFe$_2$As$_2$ (see Supplementary information for additional data and analysis).

Figure 4a, b shows $\theta/2\theta$ scans of $\sigma_z^{SF}$ around $\mathbf{Q_1} = (1, 0, 1)$ and $\mathbf{Q_2} = (1, 0, 3)$, respectively, at different temperatures. Since $\sigma_z^{SF}$ at these two wave vectors probes different combinations of $M_a$ and $M_c$, one can obtain magnitudes of the static ordered moments along the $a$-axis and $c$-axis directions at these temperatures. Figure 4c shows the temperature dependencies of the full-width-at-half-maximum (FWHM) of these peaks, indicating that the spin–spin correlation lengths are instrumental resolution limited and temperature independent. Figure 4d plots the magnetic scattering intensity ratio between (1, 0, 1) ($I_{101}$) and (1, 0, 3) ($I_{103}$), which measures the relative strength of $M_c$ and reveals a clear peak around $T_N/T_s$.

To further determine the effect of uniaxial pressure on $c$-axis ordered moment and its pressure dependence, we carried out unpolarized neutron diffraction measurements focusing on the magnetic scattering intensity ratio between (1, 0, 1) ($I_{101}$) and (1, 0, 3) ($I_{103}$) using an in-situ uniaxial pressure device. Since our polarized neutron diffraction measurements revealed no ordered moment $M_b$, we used unpolarized neutron diffraction on BT-7 to improve the statistics of the data across $T_N$. Figure 4e compares the measured $I_{101}/I_{103}$ from 130 to 150 K at $P \approx 0$ and 45 MPa uniaxial pressure. Consistent with earlier work[6,7], $I_{101}/I_{103}$ is approximately temperature-independent across $T_N$ at $P \approx 0$, thus indicating that the internal strain of the system does not induce a $c$-axis ordered moment. Upon applying an uniaxial pressure of $P \approx 45$ MPa, the identical measurement shows a dramatic peak at $T_N$, thus confirming the results of Fig. 4a–d. Figure 4f shows the uniaxial pressure dependence of the measured $M_c/M_a$ at $T_N$, suggesting that the ordered $c$-axis moment saturates with increasing pressure above 45 MPa.

Figure 1f shows the temperature dependencies of the magnetically ordered moments along the $a$-axis ($M_a$) and $c$-axis ($M_c$) directions obtained by comparing $\sigma_x^{SF}$ and $\sigma_z^{SF}$ at wave vectors $\mathbf{Q_1} = (1, 0, 1)$ and $\mathbf{Q_2} = (1, 0, 3)$ (see Supplementary information for additional data and analysis). In the low-temperature AF ordered state, the strain-free and strained BaFe$_2$As$_2$ have the standard collinear AF structure with no evidence of $M_c$ (right panel in Figs. 1e and 4e)[6,7]. On warming to 143 K just below $T_N$, the easy axis tilts from the $a$-axis toward the $c$-axis with an angle of ~28° (middle panel in Fig. 1e). Finally, on warming to temperatures well above $T_N$, there is no static AF order (left panel in Fig. 1e). Figure 1g shows the temperature dependence of $M_c$ at ~20 (blue solid line) and ~45 (pink solid line) MPa uniaxial pressure, compared with the uniaxial strain-induced lattice distortion $\delta(P \approx 20$ MPa$) - \delta(P = 0)$ (green solid circles and lines) obtained from neutron Larmor diffraction experiments[27]. The similarity of the data suggests that the $c$-axis aligned magnetic moment arises from the uniaxial pressure-induced lattice distortion.

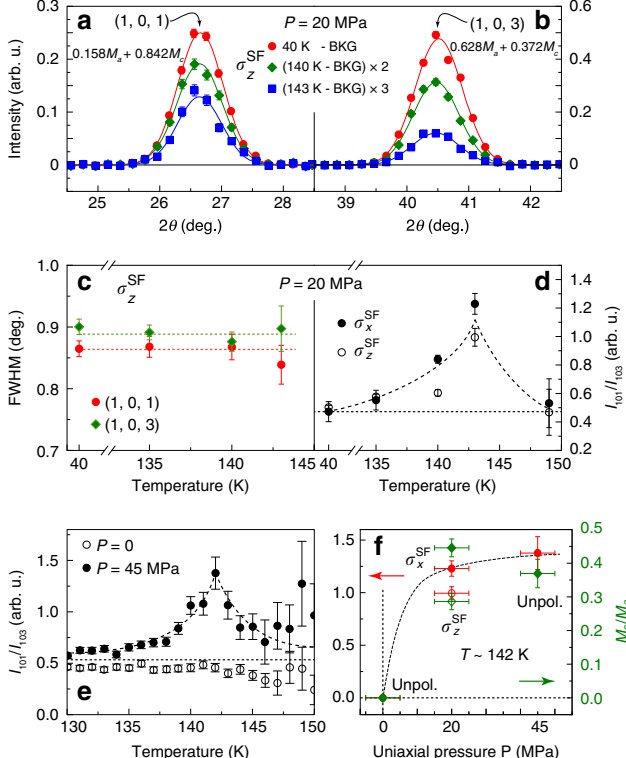

**Fig. 4 Uniaxial pressure dependence of the magnetic order and correlations.** Elastic $\theta/2\theta$ scans of $\sigma_z^{SF}$ across **a** $\mathbf{Q_1} = (1, 0, 1)$ and **b** $\mathbf{Q_2} = (1, 0, 3)$ at different temperatures and $P = 20$ MPa. The data are collected on BT-7 using final neutron energy of 14.7 meV with instrumental energy resolution of about 1.3 meV. Similar scans of $\sigma_x^{SF}$ are discussed in Supplementary information for additional data and analysis. $\sigma_y^{SF}$ was unavailable at the time of this experiment. **c** FWHM of the elastic (1, 0, 1) and (1, 0, 3) scans across $T_N/T_s$. **d** Temperature dependence of $I_{101}/I_{103} \propto (0.16 M_a + 0.84 M_c)/(0.63 M_a + 0.37 M_c)$. **e** Temperature dependence of $I_{101}/I_{103}$ at $P \approx 0$ and ~ 45 MPa uniaxial pressure obtained using in-situ uniaxial pressure device with unpolarized neutrons on BT-7. **f** Pressure dependence of $I_{101}/I_{103}$ (red symbols, left axis) and $M_c/M_a$ (green symbols, right axis) at $T \sim 142$ K. The data points for $P = 0$ and 45 MPa were measured with unpolarized neutron scattering. The data points for $P = 20$ Mpa were measured with polarized mode with the open (solid) symbols obtained from $\sigma_z^{SF}$ ($\sigma_x^{SF}$). The black dashed curve is a guide to the eye for the data points. unpol. denotes unpolarized neutron scattering measurements. The vertical error bars in (**a**, **b**, **d**, **e**) represent statistical errors of 1 standard deviation. The error bars in (**c**) are estimated errors from fits to magnetic Bragg peak widths. The error bars in (**f**) are our estimated errors from fits to magnetic order parameters and applied uniaxial pressure.

## Discussion

Theoretically, the in-plane electronic anisotropy of the iron pnictides is expected to couple linearly to the lattice orthorhombicity by the Ginzburg–Landau free-energy formalism if one ignores the effect of spin-orbit coupling induced magnetic anisotropy[9,27]. From this perspective, in-plane uniaxial strain should only induce in-plane electronic anisotropy. The discovery of a $c$-axis ordered magnetic moment coupled exclusively with uniaxial pressure-induced lattice distortion suggests that such an effect cannot be only associated with the lattice orthorhombicity of the system, as $M_c$ becomes vanishingly small in the low-temperature orthorhombic phase with large in-plane lattice distortion. This is also different from the $c$-axis moment AF structure in $Ba_{1-x}K_xFe_2As_2$ in the sense that the $c$-axis order appears

exclusively in the tetragonal phase[42,43], while the $c$-axis moment appears in $BaFe_2As_2$ only near the peak of the nematic susceptibility around $T_N/T_s$. Although there is currently no theoretical understanding of this observation, it must arise from spin-orbit coupling induced magnetic anisotropy[44]. Our discovery opens a new avenue to control magnetic order in nematic materials using mechanical strain instead of magnetic fields. The strong coupling of the $c$-axis aligned magnetic order with an in-plane pressure-induced lattice distortion offers the potential for the next generation of mechanical-strain-controlled magnetic switches. One must consider the presence of the magnetically ordered moment along the $c$-axis in mechanically detwinned iron pnictides in order to understand their intrinsic electronic, magnetic, and nematic properties.

Alternatively, our observations are also consistent with strain inducing a proximate $XY$ spin anisotropy near $T_N/T_s$. In this scenario, while $a$-axis is energetically favorable in terms of spin anisotropy, $c$-axis is very close in energy. This allows for a distribution of large (resolution-limited but not long-range ordered) and long-lived (quasi-static) collinear magnetic domains, with their collinear spin direction in the $ac$-plane. The ratio between $I_{101}/I_{103}$ (Fig. 4d, e) is then a measure of the distribution of domains, reflective of the difference in spin anisotropy energies along the $a$- and $c$-axis. Similar to when the easy axis tilts from $a$-axis toward $c$-axis under strain (Fig. 1e), the change to a proximate $XY$ spin anisotropy under strain also indicates a large and highly unusual effect of strain on the spin anisotropy.

In conclusion, we have used polarized and unpolarized neutron scattering to study the magnetic structure and critical scattering in uniaxial strained $BaFe_2As_2$. We find that the uniaxial pressure necessary to make single domain samples of $BaFe_2As_2$ also induces $c$-axis polarized critical magnetic scattering and static magnetic order around $T_N/T_s$. The size of the $c$-axis ordered moment is associated with the uniaxial pressure-induced lattice distortion, instead of the lattice orthorhombicity. These results indicate that in addition to detwinning $BaFe_2As_2$, uniaxial pressure applied on the sample actually modifies the magnetic structure of the system. Therefore, infrared[45], angle resolved photoemission[16], and Raman spectroscopy[46,47] experiments on mechanically detwinned $BaFe_2As_2$ near the magnetic and nematic phases should be reexamined to take into account the effect of strain-induced change to the spin anisotropy on the in-plane electronic and magnetic properties.

## Methods

**Sample preparation and experimental details.** $BaFe_2As_2$ single crystals were grown by the self-flux method using the same growth procedure as described before[19]. Our polarized inelastic neutron scattering experiments were carried out using the CEA CRG-IN22 triple-axis spectrometer at the Institut Laue-Langevin, Grenoble, France (All raw data from ILL will be published at https://doi.org/10.5291/ILL-DATA.4-02-531, and from NCNR will be provided upon request). Polarized neutrons were produced using a focusing Heusler monochromator and analyzed with a focusing Heusler analyzer with a final wave vector of $k_f = 2.662$ Å$^{-1}$. The experimental setups for uniaxial pressured and pressure freed experiments are identical. However, it is difficult to directly compare the scattering intensity of these two experiments since the sample masses, their relative positions in the beam, and background scattering of these two experiments are different. Nevertheless, one can safely compare the relative intensity changes of these two experiments. The polarized elastic neutron scattering experiments were carried out on BT-7 utilizing $^3$He polarizers immediately before and after the sample at NIST center for neutron research, Gaithersburg, Maryland, USA[40,41]. The unpolarized neutron diffraction experiments in Fig. 4e were carried out using a pyrolytic graphite monochromator and analyzer with pyrolytic graphite filter in the beam. Experiments on twinned $BaFe_2As_2$ without external uniaxial pressure were performed on ~12-g aligned single crystals as described before[34]. The polarized inelastic neutron scattering experiments on uniaxial pressured detwinned $BaFe_2As_2$ were performed using 12 pieces cut single crystals (~3 g, Fig. S1)[21]. The BT-7 measurements were carried out on a single piece of $BaFe_2As_2$ mounted on a newly built in-situ uniaxial pressure device and the neutron wave vectors are set at $k_i = k_f = 2.662$ Å$^{-1}$.

**Determination of $M_a$, $M_b$, and $M_c$.** In our previous polarized neutron scattering studies of iron pnictides, we have established the method for determining the spin-fluctuation components $M_\beta$ ($\beta = a, b, c$) along the lattice axes via comparing the SF scattering $\sigma_\gamma^{SF}$ ($\gamma = x, y, z$) at two equivalent magnetic wave vectors (such as $\mathbf{Q}_1 = (1, 0, 1)$ and $\mathbf{Q}_2 = (1, 0, 3)$ as shown in Fig. 1 of the main text). The definition of the directions $x$, $y$, and $z$ is described in Fig. 1. $\sigma_\gamma^{SF}$ is directly related to the spin-fluctuation components by:

$$
\begin{cases}
\sigma_x^{SF}(\mathbf{Q}) = F^2(\mathbf{Q})\sin^2\alpha_\mathbf{Q}\frac{R}{R+1}M_a + F^2(\mathbf{Q})\frac{R}{R+1}M_b + F^2(\mathbf{Q})\cos^2\alpha_\mathbf{Q}\frac{R}{R+1}M_c + B(\mathbf{Q}),\\
\sigma_y^{SF}(\mathbf{Q}) = F^2(\mathbf{Q})\sin^2\alpha_\mathbf{Q}\frac{1}{R+1}M_a + F^2(\mathbf{Q})\frac{R}{R+1}M_b + F^2(\mathbf{Q})\cos^2\alpha_\mathbf{Q}\frac{1}{R+1}M_c + B(\mathbf{Q}),\\
\sigma_z^{SF}(\mathbf{Q}) = F^2(\mathbf{Q})\sin^2\alpha_\mathbf{Q}\frac{R}{R+1}M_a + F^2(\mathbf{Q})\frac{1}{R+1}M_b + F^2(\mathbf{Q})\cos^2\alpha_\mathbf{Q}\frac{R}{R+1}M_c + B(\mathbf{Q}),
\end{cases}
\tag{1}
$$

where $\alpha$ is the angle between $(1, 0, 0)$ and $\mathbf{Q}$ (Fig. 1), $F(\mathbf{Q})$ is magnetic form factor of $Fe^{2+}$, $R$ is the flipping ratio ($R = \sigma_{Bragg}^{NSF}/\sigma_{Bragg}^{SF} \approx 13$), and $B$ is the polarization-independent background scattering. From Eq. (1), we can get four equations for our results on $\mathbf{Q}_1$ and $\mathbf{Q}_2$:

$$
\begin{cases}
\sigma_x^{SF}(\mathbf{Q}_1) - \sigma_y^{SF}(\mathbf{Q}_1) = \frac{R-1}{R+1}F^2(\mathbf{Q}_1)[\sin^2\alpha_1 M_a + \cos^2\alpha_1 M_c],\\
\sigma_x^{SF}(\mathbf{Q}_2) - \sigma_y^{SF}(\mathbf{Q}_2) = r\frac{R-1}{R+1}F^2(\mathbf{Q}_2)[\sin^2\alpha_2 M_a + \cos^2\alpha_2 M_c],\\
\sigma_x^{SF}(\mathbf{Q}_1) - \sigma_z^{SF}(\mathbf{Q}_1) = \frac{R-1}{R+1}F^2(\mathbf{Q}_1)M_b,\\
\sigma_x^{SF}(\mathbf{Q}_2) - \sigma_z^{SF}(\mathbf{Q}_2) = r\frac{R-1}{R+1}F^2(\mathbf{Q}_2)M_b,
\end{cases}
\tag{2}
$$

in which $r$ is the intensity ratio factor between $\mathbf{Q}_1$ and $\mathbf{Q}_2$ to account for the differences in sample illumination volume and the convolution with instrumental resolution. The third and fourth equations in Eq. (2) can be used to determine the ratio $r$ and $M_b$, and the first two equations for $M_a$ and $M_c$. More details concerning the determination of the spin-fluctuation components $M_a$, $M_b$, and $M_c$ can be found elsewhere[39]. Although this method can determine the values of $M_a$, $M_b$, and $M_c$, it also results in large error bars of their values. To more accurately determine the effect of uniaxial pressure on $M_a$ and $M_c$, we consider the differences between $\sigma_z^{SF}(\mathbf{Q}) - \sigma_y^{SF}(\mathbf{Q})$ at $\mathbf{Q}_1$ and $\mathbf{Q}_2$.

$$
\begin{cases}
\sigma_z^{SF}(\mathbf{Q}_1) - \sigma_y^{SF}(\mathbf{Q}_1) = \frac{R-1}{R+1}F^2(\mathbf{Q}_1)[\sin^2\alpha_1 M_a + \cos^2\alpha_1 M_c - M_b] \propto 0.16M_a + 0.84M_c - M_b\\
\sigma_z^{SF}(\mathbf{Q}_2) - \sigma_y^{SF}(\mathbf{Q}_2) = r\frac{R-1}{R+1}F^2(\mathbf{Q}_2)[\sin^2\alpha_2 M_a + \cos^2\alpha_2 M_c - M_b] \propto 0.63M_a + 0.37M_c - M_b
\end{cases}
\tag{3}
$$

As $M_b$ does not diverge in uniaxial pressured and pressure-free cases[26], a comparison of $\sigma_z^{SF}(\mathbf{Q}_1) - \sigma_y^{SF}(\mathbf{Q}_1)$ raw data should be most sensitive to changes in $M_c$, while $\sigma_z^{SF}(\mathbf{Q}_2) - \sigma_y^{SF}(\mathbf{Q}_2)$ should be sensitive to changes in both $M_a$ and $M_c$. The outcome of this analysis is shown in Figs. 2e, f and 3e, f.

In our polarized neutron diffraction experiment at BT-7, we have only measured $\sigma_x^{SF}$ and $\sigma_z^{SF}$. In elastic channel, $M_\beta$ is proportional to the square of the ordered moment ($m_\beta$). The determination of $M_\beta$ follows the same method as described in Eqs. (1) and (2). But we need to apply the Lorentz factor ($L = \frac{1}{\sin 2\theta}$) as we use the integrated intensity of $\theta - 2\theta$ scan to calculate $M_\beta$[48], where $2\theta$ is the scattering angle for $\mathbf{Q}$. Moreover, since no divergence of critical spin fluctuations were observed along the $b$-axis, we can assume the absence of static ordered moment ($M_b = 0$) (even if we consider that quasi-elastic spin fluctuations along $b$-axis within the energy resolution of the elastic scattering could be included in $\sigma_x^{SF}$ and $\sigma_z^{SF}$, it can be neglected at least in $\sigma_z^{SF}$ because of the small pre-factor $\frac{1}{R+1} \approx 0.07$ before $M_b$). Then Eq. (2) can be written as:

$$
\begin{cases}
\sigma_x^{SF}(\mathbf{Q}_1) = \sigma_z^{SF}(\mathbf{Q}_1) = \frac{1}{\sin 2\theta_1}\frac{R}{R+1}F^2(\mathbf{Q}_1)[\sin^2\alpha_1 M_a + \cos^2\alpha_1 M_c]\\
\sigma_x^{SF}(\mathbf{Q}_2) = \sigma_z^{SF}(\mathbf{Q}_2) = r\frac{1}{\sin 2\theta_2}\frac{R}{R+1}F^2(\mathbf{Q}_2)[\sin^2\alpha_2 M_a + \cos^2\alpha_2 M_c]
\end{cases}
\tag{4}
$$

Given the magnetic moment is polarized along $a$-axis at 40 K $\ll T_N$ with $m_a \approx 0.87\mu_B$, we can get $r$, solve $M_a$ and $M_c$ from both $\sigma_x^{SF}$ and $\sigma_z^{SF}$, and determine the magnitude of the $c$-axis moment induced by uniaxial strain. Taking $m_a = 0.87\mu_B$ at 40 K, we can get $m_a$ and $m_c$ at other temperatures using the data points shown in Fig. 1f. From $\sigma_z^{SF}$, we get $m_a \approx 0.23 \pm 0.05\mu_B$ and $m_c \approx 0.12 \pm 0.03\mu_B$ at 143 K, resulting in a canting angle of ~28° at this critical temperature. The calculated canting angles are estimated to be about 14° at 140 and 149 K, and gradually decrease to zero below 135 K.

**$\sigma_{x,y,z}^{SF}$ and $M_{a,b,c}$ below and well above $T_N$ at the AF ordering wave vectors.** Figure S2 shows the results of $\sigma_\gamma^{SF}$ ($\gamma = x, y, z$) below and well above $T_N$ under zero and $P \sim 20$ MPa. At $T = 135$ K ($< T_N$), $\sigma_\gamma^{SF}$ for uniaxial pressure-free and pressured cases are shown in Fig. S2a–d. A comparison of $\sigma_z^{SF}(\mathbf{Q}_1) - \sigma_y^{SF}(\mathbf{Q}_1)$ scattering at $P = 0$ and ~20 MPa in Fig. S2e suggests that the applied uniaxial pressure may enhance $M_c$ around ~8 meV. Similar data at $\mathbf{Q}_2$ in Fig. S2f suggest that the effect of uniaxial pressure is limited on $M_a$ at this temperature. Figure S2g, h shows as the converted $M_a$, $M_b$, and $M_c$ at $T = 135$ K. At $T < T_N$, the data with $P \sim 20$ MPa are qualitatively consistent with that measured on the $P = 0$ sample, except that both the $M_a$ and $M_b$ are gapped below $E > 10$ meV and ~6 meV, respectively, while only $M_a$ is gapped below 6 meV for the $P = 0$ sample. Note $T_N$ is ~136 K for $P = 0$ and ~143 K for $P \sim 20$ MPa. In relative temperature $T/T_N$, 135 K is much lower in the $P \sim 20$ MPa sample ($0.94T_N$) than that in free-standing sample ($0.99T_N$), thus the

spin fluctuations are further gapped. For temperatures well above $T_N$ (Fig. S2i–p), SF scattering becomes very weak and no qualitative difference was observed for $P = 0$ and $P \sim 20$ MPa.

**Comparison of $M_\beta$ at $\mathbf{Q} = (1, 0)$ and $(0, 1)$.** To determine if the uniaxial pressure-induced $M_c$ at the AF wave vector $\mathbf{Q} = (1, 0)$ is compensated by magnetic scattering reduction at $(0, 1)$, we compare $\sigma_\gamma^{SF}$ between $\mathbf{Q} = (1, 0, L)$ and $(0, 1, L)(L = 1, 3)$ at $T = 145$ K (Fig. S3a–d). Figure S3e, f shows the energy dependence of $\sigma_z^{SF}(\mathbf{Q}) - \sigma_y^{SF}(\mathbf{Q})$ at $\mathbf{Q} = (1, 0, 1)/(0, 1, 1)$ and $\mathbf{Q} = (1, 0, 3)/(0, 1, 3)$. Compared with clear magnetic intensity gains below ~6 meV at the AF wave vectors $\mathbf{Q}_1 = (1, 0, 1)$ and $\mathbf{Q}_2 = (1, 0, 3)$, paramagnetic scattering at $\mathbf{Q} = (0, 1, 1)$ and $(0, 1, 3)$ is isotropic in spin space as illustrated by the zero values of $\sigma_z^{SF}(\mathbf{Q}) - \sigma_y^{SF}(\mathbf{Q})$ at these wave vectors. Figure S3g, h shows the energy dependence of $M_a$, $M_b$, and $M_c$ extracted from Fig. S3a–d at the wave vectors $(1, 0)$ and $(0, 1)$, respectively. Therefore, the applied uniaxial pressure clearly has an impact on magnetic excitations at $(1, 0)$ but has no observable effect at $(0, 1)$, which has weak and featureless energy dependence of isotropic $M_a$, $M_b$, and $M_c$ (Fig. S3h).

Consistent with the weak scattering at $(0, 1, L)$ observed at 145 K, temperature dependence of $M_a$, $M_b$, and $M_c$ at $\mathbf{Q} = (0, 1)$ is much weaker than that at $(1, 0, L)$ and decreases in intensity at $T_N$ (Fig. S4), consistent with the temperature dependence of $(0, 1, 1)$ in detwinned $BaFe_2As_2$ measured with unpolarized neutron scattering[21].

**Uniaxial pressure dependence of the magnetic order and correlations.** Figure S5 summarizes the elastic $\theta - 2\theta$ scans of $\sigma_x^{SF}$ across $\mathbf{Q} = (1, 0, L)(L = 1, 2, 3)$. Similar to the $\theta - 2\theta$ scans of $\sigma_z^{SF}$ as described in Fig. 4 of the main text, the scans for $\sigma_x^{SF}$ (Fig. S5a, b) exhibit temperature-independent FWHM from 40 to 143 K (Fig. S5c), indicating that the spin–spin correlation length is resolution limited even in the temperature range above $T_N \sim 136$ K of unstrained sample. Figure S5d plots the ratio between the scattering intensity at $(1, 0, 1)$ and $(1, 0, 3)$ ($I_{101}/I_{103}$), which is greatly enhanced close to $T_N$. Since $\sigma_x^{SF} = 0.16M_a + 0.84M_c$ at $\mathbf{Q}_1 = (1, 0, 1)$ and $0.37M_a + 0.37M_c$ at $\mathbf{Q}_2 = (1, 0, 3)$, the enhancement of $I_{101}/I_{103}$ is consistent with the emergence of a $c$-axis magnetic moment induced by uniaxial strain. At temperature where $M_c$ is not induced, the ratio $I_{101}/I_{103} = 0.16M_a/0.63M_a \times \frac{\sin^2 2\theta_2}{\sin^2 2\theta_1} \approx 0.5$ (black dashed line in Fig. S5d), where $\frac{\sin^2 2\theta_2}{\sin^2 2\theta_1}$ accounts for the Lorentz factor. The data points of $I_{101}/I_{103}$ in Fig. S5d show that $M_c$ is absent at 149 K and below 135 K but reaches a maximum at 143 K close to $T_N$. The unpolarized data in Fig. 4f show similar behavior.

In addition to the emergence of $M_c$, it is also important to understand whether $M_c$ forms a new periodicity along $c$-axis. The magnetic structure factor of the three-dimensional AF order of $BaFe_2As_2$ results in magnetic peaks at $(1, 0, L)$ with $L = 1, 3, 5, \dots$ and the absence of magnetic scattering at $(1, 0, L)$ with $L = 0, 2, 4, \dots$. If the induced $M_c$ forms a larger magnetic unit cell along $c$-axis that ensures the presence of $(1, 0, 1)$ and $(1, 0, 3)$, one can expect detectable magnetic scattering at $L = 0, 2$. However, the three-point $\theta - 2\theta$ across $(1, 0, 2)$ in Fig. S5 shows that the intensity for $(1, 0, 2)$ is smaller than 1/3000 of $(1, 0, 3)$, which rules out this possibility and further confirms our conclusion about the canting-moment picture as shown in Fig. 1 of the main text.

## Data availability

The data that support the findings of this study are available from the corresponding authors on request, and all raw data from ILL will be published at https://doi.org/10.5291/ILL-DATA.4-02-531, and from NCNR will be provided upon request.

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

## Acknowledgements

The work at BNU is supported by the NSFC under Grant No. 11922402 and 11734002 (X.L.). The work at Rice University is supported by the U.S. NSF DMR-1700081 and the Robert A. Welch Foundation Grant No. C-1839 (P.D.). The work at UCB was supported by the U.S. DOE BES under Contract No. DE-AC02-05-CH11231 within the Quantum Materials Program (KC2202).

## Author contributions

P.D. and X.L. conceived the project. P.L., M.L.K., L.T., G.T. and X.L. prepared the samples. Polarized inelastic neutron scattering experiments at IN22 were carried out by P.L., L.T., X.L., K.S., J.T.P., Y.L., Y.X.S., and F.B. Neutron diffraction measurements at BT-7 were carried out by M.L.K, Y.S., D.W.T., J.W.L, Y.Z., and R.J.B. The entire project was supervised by P.D. The paper was written by P.D., P.L., X.L., and Y.S. All authors made comments.

## Competing interests

The authors declare no competing interests.
