## [Peer Review File · Nature Communications]

Reviewers' Comments:

Reviewer #1:

Remarks to the Author:

The paper by Liu et al. reports the observation of out-of-plane magnetic moments in BaFe₂As₂ induced by uniaxial stress. This conclusion is based on the results of polarized neutron-scattering measurements. While I find this result sufficiently intriguing, I am not convinced by the quality of the presented data and doubt the statistical significance of such a conclusion. I also see significant gaps in the presentation of the measurement results and their interpretation, as I explain below.

In Fig. 2, the authors conclude that the M_c component in the paramagnetic state is significantly larger in the strained sample as compared to the reference measurement with $P=0$. This conclusion is effectively based on a single data point at $E = 2$ meV (blue data set), which happens to be higher in panel (f) than in panel (e). I find the way how the data in these figures is plotted very misleading, because the two panels have a different vertical scale, which is in addition offset in panel (e), providing the impression that the data lie at the same level at high energies, which is in fact not true. At high energies, the data in panel (f) level off at ~ 190 counts/10min, whereas in panel (e) the same level is ~ 90 counts/10 min, so it is not clear how these two datasets are to be compared at all. Moreover, the relative increase in the data (f) at low energies (which is artificially highlighted by the "guide to the eyes") is of the order of 100 counts/10 min. This is not larger than the deviation of the rightmost red data point in panel (e) from the solid line! The authors nevertheless choose to emphasize the low-energy feature with the "guide to the eyes", neglecting the higher-energy point as a statistical outlier (which it probably is). To report a convincing effect, I would expect that it shows up with statistical significance in several neighboring points in a certain energy range, and not just for one point.

One sees similar problems with the data interpretation in Fig. 3. The "guides to the eyes" appear to indicate that the blue curve in panel (f) is significantly different from that in panel (e). However, the data point in (f) that is located exactly at T_N (on the dashed vertical line, where the signal should diverge) coincides with the green dataset! Nevertheless, the authors choose to draw the solid lines in such a way that the cusps are located at different positions for every polarization direction, which artificially exaggerates the desired effect. To my understanding, the cusps in the curves should be all located at T_N , which is obviously independent of the polarization. For an unbiased analysis, one should introduce some sort of an empirical fitting function to model this cusp and then fit it globally to all the measured curves, keeping T_N a common parameter for all the fits. As the authors state in the text, the existence of magnetic moments along the c -direction should lead to a divergence in the M_c curve at T_N , yet this is not observed in the data because the point right at T_N has a very low value.

How to understand the very low energy scale of ~ 4 meV, below which the enhancement of fluctuations (e.g. in Fig. 2f) occurs? This energy is much smaller than the exchange constants and even smaller than the anisotropy energy, which is of the order of 10 meV in this compound. What should forbid the out-of-plane magnetic moments to fluctuate faster? It is also not clear, where the spectral weight for the corresponding enhancement comes from. Does it imply that there should be a similar depletion of the spectral weight at the orthogonal wave vector? It would be also natural to see how the spectrum changes upon the transition into the ordered state. I saw no justification in the text, why the inelastic data are limited only to the paramagnetic state.

If the claimed effect is indeed real, how does it depend on the applied stress? In a twinned, multi-domain sample, local uniaxial stresses exist naturally, exerted on every tetragonal grain from the neighboring grains with a different orientation of the a/b axes. Does this mean that out-of-plane magnetic moments should be also expected in as-grown crystals that did not undergo any detwinning procedure? Or does the stress have a certain threshold value below which the effect is zero?

The paper is also insufficiently clear about the experimental protocol. In particular, there is no information how the reference $P=0$ state was measured. Was it exactly the same sample before tightening the screws on the detwinning device? Apparently, the design of the sample holder does not allow to detwin the sample in-situ and then remove the uniaxial pressure to achieve the single-domain magnetic state with no applied stress. This means, the $P=0$ state was probably the multi-domain state that has some finite distribution of uniaxial strains and is not at all strain-free, contrary to what the authors claim. The fact that the sample experiences no pressure as a whole does not mean that the individual grains do not exert such pressure on each other.

In view of these weaknesses, I cannot recommend this version of the paper for publication. While there is indeed some weak evidence in the data suggestive of the reported effect, these experiments can only be seen as preliminary measurements to justify future investigations, not at all as a completed work of publishable quality. The only way how this paper can be made convincing is by performing additional measurements, where the effect appears not just sporadically in a couple of data points, but can be seen clearly, reproducibly, and with high statistical significance in a well defined range of energies and at least several uniaxial pressures.

Reviewer #2:

Remarks to the Author:

The current manuscript reports a polarized neutron diffraction study of single crystals of BaFe_2As_2 under in-plane uniaxial pressure. The BaFe_2As_2 hosts a colinear AFM order with moment along the orthorhombic a -axis. The authors found that the sample under pressure shows enhanced critical fluctuations of moments along the c -axis. The static moment also shows a tilting toward c -axis near T_N/T_S . All these findings are consistent with previous NMR work (Nat. Comm. 9, 1058). The out-of-plane character in the spin space of nematic phase of iron pnictides is a very interesting and surprising finding. This work is a nice complementary study of the previous NMR paper. Since the previous NMR work is published in Nat. Comm, I believe this paper is also suitable for the same journal. Below are some minor comments/suggestions for the authors:

1. This measurement is measured under constant pressure, whereas the NMR paper is measured under in-situ tunable strain. Is it possible to compare the results of the two works and see if there is a quantitative agreement? For example, plotting M_c / Δ , where Δ is the measured orthorhombicity in Fig. 1, and compare it with χ^{nem}_{zz} in fig. 4 of the NMR paper?
2. The in-plane uni-axial pressure not only induces orthorhombicity, but also likely induces a change of tetragonality (c/a) ratio, which might also affect the magnetic anisotropy. Since the elastic modulus softens as temperature approaches T_S , this effect may also enhance at T_S . Does this effect also contribute to the observed phenomenon? This has not been addressed in the NMR papers. Can the authors comment on this based on their structural data?
3. A minor typo: In the last paragraph of page 3. There is a sentence "Figures 2(c), 2(d) and 2(f) show identical scans as those of Fig. 1(a), 1(b) and 1(e), respectively." I believe it should be ""Figures 2(c), 2(d) and 2(f) show identical scans as those of Fig. 2(a), 2(b) and 2(e), respectively.""

Reviewer #3:

Remarks to the Author:

The present paper entitled 'In-plane uniaxial pressure-induced out-of-plane antiferromagnetic moment and critical fluctuations in BaFe_2As_2 ' is a polarised neutron scattering study on the aforementioned iron pnictide. Thanks to this technique, the authors are able to extract the different components of the magnetic moment, along the different axis directions of the crystal. The main goal of the study is to provide evidence for the existence of an out-of-plane (along c -axis) antiferromagnetic moment when applying in-plane uniaxial pressure. This would be consistent with previous NMR studies, and above all it is expected to give new insight into the

impact of pressure on these iron-based superconductors, which should be reconsidered in previous and future studies. This is potentially important for the understanding of these systems, and I believe the work deserves publication in general.

However although the present data contribute to an exciting debate, the extracted information, its statistics and guiding curves are not enough convincing. The magnitude of the out-of-plane moment, if existing, could be even weaker than proposed, and then it could be considered as a negligible effect.

If one considers the figure 2, which compares the energy-dependence of the spin-flip cross-sections and extracted magnetic moment components with and without applying pressure, it is indeed showing, at least from the raw data, there is an effect of pressure. Nevertheless, when considering the extracted magnetic moment components, and especially the guides to the eye, the effect and the amplitudes at low energy are overestimated. Further justification and discussion on these results would be necessary.

Looking now at figure 3, which is proposed to demonstrate further, I'm actually less convinced. It compares the temperature-dependence with and without pressure for the same quantities as in figure 2, at 2meV where the effect is maximum. The effect of pressure is less obvious and the guide to the eye for the magnetic moment along c when applying pressure is misleading. The peak shape actually relies on only one point at 145K. M_c does not clearly diverges around T_N/T_s , especially if one considers the point around 135K which is also out of the curve.

Concerning figure 4 and especially figure 4d, a note clarifying why it corresponds to M_c should be added. It would be more convincing with more points, more temperatures, here as well.

Finally, large scale facilities nowadays provides DOI for experimental data, the authors could add its reference.

Reviewers' comments:

Reviewer #1 (Remarks to the Author):

"The paper by Liu et al. reports the observation of out-of-plane magnetic moments in BaFe₂As₂ induced by uniaxial stress. This conclusion is based on the results of polarized neutron-scattering measurements. While I find this result sufficiently intriguing, I am not convinced by the quality of the presented data and doubt the statistical significance of such a conclusion. I also see significant gaps in the presentation of the measurement results and their interpretation, as I explain below."

We appreciate these comments from the referee.

"In Fig. 2, the authors conclude that the M_c component in the paramagnetic state is significantly larger in the strained sample as compared to the reference measurement with P=0. This conclusion is effectively based on a single data point at E = 2 meV (blue data set), which happens to be higher in panel (f) than in panel (e). I find the way how the data in these figures is plotted very misleading, because the two panels have a different vertical scale, which is in addition offset in panel (e), providing the impression that the data lie at the same level at high energies, which is in fact not true. At high energies, the data in panel (f) level off at ~190 counts/10min, whereas in panel (e) the same level is ~90 counts/10 min, so it is not clear how these two datasets are to be compared at all. Moreover, the relative increase in the data (f) at low energies (which is artificially highlighted by the "guide to the eyes") is of the order of 100 counts/10 min. This is not larger than the deviation of the rightmost red data point in panel (e) from the solid line! The authors nevertheless choose to emphasize the low-energy feature with the "guide to the eyes", neglecting the higher-energy point as a statistical outlier (which it probably is). To report a convincing effect, I would expect that it shows up with statistical significance in several neighboring points in a certain energy range, and not just for one point."

We appreciate very much these instructive comments. The experiments in zero pressure were carried out twinned Ba₁₂₂ and the sample mass is different from experiments on uniaxial pressured Ba₁₂₂. Therefore, it is difficult to directly compare the intensity of these two experiments. One of the key features of a polarized neutron scattering experiment is that one can obtain the magnitude of the magnetic scattering without having to worry about the background. Therefore, the scattering intensity in Figs. 2(e) and 2(f) are pure estimated magnetic intensity. As discussed in the SI of the manuscript, the way we calculate the c-axis moment is to compare intensity of the scattering at different L-values, which introduces a lot of uncertainty in calculated value of M_c, M_a, and M_b. The best way to directly determine the effect of an in-plane uniaxial pressure is to compare the raw data directly for zero and finite pressure cases. In polarized neutron scattering, $\sigma_{xx} - \sigma_{yy} = M_y \sin^2 \alpha + M_c \cos^2 \alpha$; $\sigma_{zz} = M_y$; and $\sigma_{zz} - \sigma_{yy} = M_y - M_z$; and $M_z = M_b$. Therefore, by using $\sigma_{zz} - \sigma_{yy}$, we are essentially probing $M_y - M_z$, which is the same as M_y as M_b is zero before and after the pressure.

The magnetic Bragg peak (1,0,1) has $\alpha = 23$ degree, giving $M_y \approx 0.16M_a + 0.84M_c$. Similarly, the magnetic Bragg peak (1,0,3) has $\alpha = 52.4$, giving $M_y = 0.63M_a + 0.37M_c$. Therefore, by comparing the pressure on and pressure of magnetic components of (1,0,1) and (1,0,3) peak, we should be able to conclusive determine the pressure induced effect on M_c. At zero pressure, (1,0,1) should not be sensitive to M_a divergence around TN, so there should be no peak around TN at low energies. This is consistent with the data. Upon applying a pressure, one can see diverging intensity, which should be mostly sensitive to diverging M_c. At (1,0,3), zero pressure should reveal diverging M_a, as clearly seen in the data, and uniaxial pressure enhances the effect due to enhanced M_c.

For isotropic paramagnetic scattering, we would expect $\chi_{yy} = \chi_{zz}$. Therefore, any differences in χ_{yy} and χ_{zz} must arise from mostly M_c as $M_z = M_b$ and has no pressure dependence (From raw data in Fig. 2(a) and 2(c)). Fig. 2(f) compares the differences $\chi_{zz} - \chi_{yy}$, indicating that pressure indeed induces magnetic anisotropy below 6 meV. With these additional calculation and estimation from the raw data, we believe a general reader will be convinced about the effect of the pressure to enhance the M_c and hope the referee will also agree.

“One sees similar problems with the data interpretation in Fig. 3. The “guides to the eyes” appear to indicate that the blue curve in panel (f) is significantly different from that in panel (e). However, the data point in (f) that is located exactly at T_N (on the dashed vertical line, where the signal should diverge) coincides with the green dataset! Nevertheless, the authors choose to draw the solid lines in such a way that the cusps are located at different positions for every polarization direction, which artificially exaggerates the desired effect. To my understanding, the cusps in the curves should be all located at T_N , which is obviously independent of the polarization. For an unbiased analysis, one should introduce some sort of an empirical fitting function to model this cusp and then fit it globally to all the measured curves, keeping T_N a common parameter for all the fits. As the authors state in the text, the existence of magnetic moments along the c-direction should lead to a divergence in the M_c curve at T_N , yet this is not observed in the data because the point right at T_N has a very low value.”

We very much appreciate these comments from the referee. To address the issue raised by the referee and determine the effect of an in-plane uniaxial pressure on temperature dependence of M_c , we carried out similar data analysis as discussed in our replies to the comments of the referees above. We also made new figures as shown in revised Figs. 3(e) and (f), which clearly demonstrate that the effect of an in-plane uniaxial pressure is to enhance ordered moment along the c-axis. The revised figure hopefully should convince the readers and the referee.

“How to understand the very low energy scale of ~ 4 meV, below which the enhancement of fluctuations (e.g. in Fig. 2f) occurs? This energy is much smaller than the exchange constants and even smaller than the anisotropy energy, which is of the order of 10 meV in this compound. What should forbid the out-of-plane magnetic moments to fluctuate faster? It is also not clear, where the spectral weight for the corresponding enhancement comes from. Does it imply that there should be a similar depletion of the spectral weight at the orthogonal wave vector? It would be also natural to see how the spectrum changes upon the transition into the ordered state. I saw no justification in the text, why the inelastic data are limited only to the paramagnetic state.”

At present, we have no theoretical understanding of the ~ 4 -6 meV energy scale of the magnetic anisotropy around T_N . However, we note that magnetic anisotropy decreases with increasing temperature. In the case of zero pressure, magnetic anisotropy occurs below about 5 meV at 138 K [Fig. 2(a), (b)]. Application of a 20 MPa does not really change the energy scale of magnetic anisotropy at 145 K [Figs. 2(c), 2(d)], and only seems to enhance c-axis spin fluctuations. The magnitude of the magnetic anisotropy is related to temperature dependence of the orbital order arising from the splitting of the d_{yz} and d_{xz} orbitals below T_s . Since the T_N of the system increases linearly with increasing uniaxial pressure [25], we expect similar splitting at 145 K in pressured sample. From this perspective, it is not surprising that the energy scale of the magnetic anisotropy is smaller near T_N compared with those at low temperature.

To address the question where the spectral weight of the diverging c-axis component comes from, we note that the ordered moment is rather a small portion of the total moment (see Fig. 1(f)). The

gained spectral weight along c-axis can come from paramagnetic background, just as the a-axis component of the spin fluctuations. We have also carried out measurements at (0,1,1) and (0,1,3) wave vector, which is orthogonal to the AF ordering wave vector, and found that is no evidence of diverging fluctuating moment (see revised SI). We also have data at low temperature AF ordered state. As these measurements are not critical to the central conclusions of the paper, we placed them in the SI.

“If the claimed effect is indeed real, how does it depend on the applied stress? In a twinned, multi-domain sample, local uniaxial stresses exist naturally, exerted on every tetragonal grain from the neighboring grains with a different orientation of the a/b axes. Does this mean that out-of-plane magnetic moments should be also expected in as-grown crystals that did not undergo any detwinning procedure? Or does the stress have a certain threshold value below which the effect is zero?”

To address this particular question from the referee, we carried out new neutron diffraction experiment. For this purpose, we used an in-situ uniaxial pressure device specially designed for neutron scattering experiments. We have originally scheduled the experiment at NIST center for neutron research in late Feb, but COVID-19 pandemic shut all neutron sources in the US. We finally were able to do this experiment in July. In the new experiment setup, we carried out two measurements, one at essentially no external pressure, and one with about 40 MPa uniaxial pressure. In the first case, we carefully measured the temperature dependence of the magnetic order parameter for (1,01) and (1,03) peaks using unpolarized neutrons. This would address the referee's first question concerning whether the effect is there without external uniaxial pressure. For easy comparison, we show in Fig. 4(f) intensity ratio of (1,0,1) and (1,0,3) as a function of temperature. At little or no pressure, we find no anomaly in (1,0,1)/(1,0,3) across T_N , thus confirming that no c-axis ordered moment from possible intrinsic strain. Upon applying a ~40 MPa uniaxial, the same ratio now shows a sharp peak at T_N [Fig. 4(f)], clearly indicating the presence a c-axis ordered moment and its temperature range. The results are consistent with our earlier work [Fig. 4(d)]. Figure 4(g) shows our estimate uniaxial pressure dependence of the effect. Although we only have three measured pressure, the results are nevertheless clearly establish the pressure dependence of the pressure-induced c-axis ordered moment.

“The paper is also insufficiently clear about the experimental protocol. In particular, there is no information how the reference $P=0$ state was measured. Was it exactly the same sample before tightening the screws on the detwinning device? Apparently, the design of the sample holder does not allow to detwin the sample in-situ and then remove the uniaxial pressure to achieve the single-domain magnetic state with no applied stress. This means, the $P=0$ state was probably the multi-domain state that has some finite distribution of uniaxial strains and is not at all strain-free, contrary to what the authors claim. The fact that the sample experiences no pressure as a whole does not mean that the individual grains do not exert such pressure on each other.”

We appreciate these comments from the referee. In the revised draft, we made absolutely clear how experiments were carried out. In the zero pressure case, the experiments were carried out on twinned samples glued on Al plates. The outcome of these experiments were published already in a previous paper. In the pressured experiments on twinned samples, new samples were prepared and cut put in detwinning device as discussed in a previous paper on spin waves. However, both zero pressure and finite pressure experiments were carried out on the same polarized triple-axis spectrometer using exactly the same setup. Therefore, one can compare the relative strength of the magnetic scattering in these two experiments, although it remains difficult to compare the scattering of these two experiments directly. In the case of twinned sample, we agree with the referee that there may still be finite distribution of intrinsic strain, but these should be averaged out and directional independent. As shown in the new data, we see no evidence of a c-axis order moment when no uniaxial pressure is applied on the sample. In the revised draft of the paper, we have made these points very clear, and also spelled out the detailed procedure of the experiments in the method

section of the paper.

"In view of these weaknesses, I cannot recommend this version of the paper for publication. While there is indeed some weak evidence in the data suggestive of the reported effect, these experiments can only be seen as preliminary measurements to justify future investigations, not at all as a completed work of publishable quality. The only way how this paper can be made convincing is by performing additional measurements, where the effect appears not just sporadically in a couple of data points, but can be seen clearly, reproducibly, and with high statistical significance in a well defined range of energies and at least several uniaxial pressures."

We appreciate these comments from the referee, and feel that improved method of analyzing the data should convince the referee about the validity of the work and its conclusion. While it is rather difficult to carry out inelastic neutron scattering experiments as a function of uniaxial pressure as these experiments will require well controlled applied pressure on multiple samples simultaneously (very difficult to do), we did carry out new diffraction experiment on a single piece of sample to study the uniaxial pressure dependence of the c-axis ordered moment. In these new measurements, only a single piece of sample is necessary for the experiment, and uniaxial pressure applied can be well controlled. The outcome of the experiment shown in revised Fig. 4(f) clearly indicates that applied uniaxial pressure increases the c-axis aligned moment that was not present at zero pressure. These measurements should convince the referee about the key conclusion of our paper.

Reviewer #2 (Remarks to the Author):

"The current manuscript reports a polarized neutron diffraction study of single crystals of BaFe₂As₂ under in-plane uniaxial pressure. The BaFe₂As₂ hosts a collinear AFM order with moment along the orthorhombic a-axis. The authors found that the sample under pressure shows enhanced critical fluctuations of moments along the c-axis. The static moment also shows a tilting toward c-axis near TN/Ts. All these findings are consistent with previous NMR work (Nat. Comm. 9, 1058). The out-of-plane character in the spin space of nematic phase of iron pnictides is a very interesting and surprising finding. This work is a nice complementary study of the previous NMR paper. Since the previous NMR work is published in Nat. Comm, I believe this paper is also suitable for the same journal. Below are some minor comments/suggestions for the authors:

1. This measurement is measured under constant pressure, whereas the NMR paper is measured under in-situ tunable strain. Is it possible to compare the results of the two works and see if there is a quantitative agreement? For example, plotting M_c / Δ , where Δ is the measured orthorhombicity in Fig. 1, and compare it with χ^{nem}_{zz} in fig. 4 of the NMR paper?"

We appreciate very much these comments from the referee, but feel that it is rather difficult to quantitatively compare the neutron data with NMR results. This is because neutron measurements were carried under a proximate ~20 MPa pressure and spin excitations are measured at finite energy, while NMR measurements were carried out under piezoelectric strain cell (which has temperature dependence) and spin fluctuations are measured at essentially $E \sim 0$ meV. Nevertheless, one can qualitatively compare these two data set. To make this more clear, we have added a dashed line in Fig. 3 of our paper to reflect the temperature dependence of the NMR signal reported in Fig. 4 of the NMR paper.

"2. The in-plane uni-axial pressure not only induces orthorhombicity, but also likely induces a change of tetragonality (c/a) ratio, which might also affect the magnetic anisotropy. Since the elastic modulus softens as temperature approaches Ts, this effect may also enhance at Ts. Does this effect also contribute to the observed phenomenon? This has not been addressed in the NMR papers. Can the

authors comment on this based on their structural data?"

The referee asked a very interesting question. In principle, the reduction in a-axis lattice should induce changes along the c-axis. However, the effect of in-plane uniaxial pressure on the c-axis lattice parameter was never measured in any experiments, and therefore it is impossible to make any solid claim on this. In the revised draft, we discuss possible effect on the uniaxial pressure on the c-axis lattice constant.

"3. A minor typo: In the last paragraph of page 3. There is a sentence "Figures 2(c), 2(d) and 2(f) show identical scans as those of Fig. 1(a), 1(b) and 1(e), respectively." I believe it should be ""Figures 2(c), 2(d) and 2(f) show identical scans as those of Fig. 2(a), 2(b) and 2(e), respectively.""

Thank you very much, this is corrected.

Reviewer #3 (Remarks to the Author):

"The present paper entitled 'In-plane uniaxial pressure-induced out-of-plane antiferromagnetic moment and critical fluctuations in BaFe₂As₂' is a polarised neutron scattering study on the aforementioned iron pnictide. Thanks to this technique, the authors are able to extract the different components of the magnetic moment, along the different axis directions of the crystal. The main goal of the study is to provide evidence for the existence of an out-of-plane (along c-axis) antiferromagnetic moment when applying in-plane uniaxial pressure. This would be consistent with previous NMR studies, and above all it is expected to give new insight into the impact of pressure on these iron-based superconductors, which should be reconsidered in previous and future studies. This is potentially important for the understanding of these systems, and I believe the work deserves publication in general.

However although the present data contribute to an exciting debate, the extracted information, its statistics and guiding curves are not enough convincing. The magnitude of the out-of-plane moment, if existing, could be even weaker than proposed, and then it could be considered as a negligible effect. If one considers the figure 2, which compares the energy-dependence of the spin-flip cross-sections and extracted magnetic moment components with and without applying pressure, it is indeed showing, at least from the raw data, there is an effect of pressure. Nevertheless, when considering the extracted magnetic moment components, and especially the guides to the eye, the effect and the amplitudes at low energy are overestimated. Further justification and discussion on these results would be necessary."

We appreciate very much these comments, which is similar to the comments of the referee 1. With the new data analysis and new plots considering only the differences between σ_{zz} - σ_{yy} , the pressure induced effect is unambiguously shown. In the revised draft, we discuss these additional analysis and spell out why we believe the new analysis should confirm our claim of c-axis uniaxial induced moment.

"Looking now at figure 3, which is proposed to demonstrate further, I'm actually less convinced. It compares the temperature-dependence with and without pressure for the same quantities as in figure 2, at 2meV where the effect is maximum. The effect of pressure is less obvious and the guide to the eye for the magnetic moment along c when applying pressure is misleading. The peak shape actually

relies on only one point at 145K. Mc does not clearly diverges around TN/Ts, especially if one considers the point around 135K which is also out of the curve."

These comments are again similar to the comments of referee 1 concerning the statistics of the data near TN. We believe the revised figure with new analysis should alleviate the concerns of the referees.

"Concerning figure 4 and especially figure 4d, a note clarifying why it corresponds to Mc should be added. It would be more convincing with more points, more temperatures, here as well.'

As discussed in our replies to referee 1, we waited about 5 months to carry out these additional measurements. Our new data shown in Figs. 4(f) and 4(g) precisely address the question raised by the referee. With these additional new data, we believe that the referee will be convinced about validity of our conclusions.

"Finally, large scale facilities nowadays provides DOI for experimental data, the authors could add its reference."

As far as we know, NIST does not have DOI for experiments carried out there yet. But ILL does have this, we have added the ILL DOI for the experiments carried out there.

Reviewers' Comments:

Reviewer #1:

Remarks to the Author:

I appreciate the efforts that the authors invested into collecting new data, revising the manuscript and explaining their conclusions in the reply. I am now more convinced that the pressure-induced effect is real. I also consider that most of my concerns raised in the first report have been resolved. The only remaining point, which is also the one raised by the third referee, is the way the authors present "guides to the eyes" to their data that are sometimes biased and overemphasize the reported effect. These issues still remain in the new manuscript and should be resolved before the paper is published.

Namely, in Fig. 2(g) the blue line is drawn below all blue (Mc) data points between 2 and 5 meV to enhance the impression that the blue and green lines coincide. Similarly, in Fig. 3(h) the peak in the blue line appears to be much sharper than the data suggest. In the middle (near T_N), it passes through the top points, completely ignoring the central point that lies much lower. On the left-hand side, however (~ 135 K), it is drawn much lower than the average, ignoring the data point at 135 K. These "tricks" in data presentation are bad practice. If there is no obvious fitting function that one can use to fit the data, one must ensure that the "guides to the eyes" that are drawn by hand pass through the middle of the dataset, reflecting the local "average" of the data points, otherwise the line is only misguiding the eyes. The authors must fix their "guides to the eyes" accordingly in these two figures.

Apart from this remark, I consider that the revised version of the paper can be accepted for publication.

Reviewer #2:

Remarks to the Author:

The authors have addressed my questions. The paper is now suitable for publication.

Reviewer #3:

Remarks to the Author:

With this revised manuscript, the authors provided efforts, being more transparent, adding details and descriptions, they even performed new measurements, their work is clearly appreciated. However, I have to say I am still not convinced by figure 3h, although I agree that in this version it is more objectively presented. The new figures, and the text improved accordingly, are helping. The general message looks more comprehensive.

I still have questions concerning the new figure 4 though, for which I had sometimes troubles to follow the description.

Are the $\theta/2\theta$ scans performed under pressure?

Looking at figure 4d, I was wondering why σ_y was not appearing. There is no mention of it in the main text but I had the answer in the SI. It has not been measured, why? I believe it would have added proof, and it is always worth checking if it is consistent with what is seen for x and z. Moreover, I found figure 4f not very clear, a more detailed description would be necessary.

Finally, here are some minor comments:

- figure 1h should be 1g, and just below in the title of the caption, Ba2As2 should be BaFe2As2
- figure 4e is supposed to show unpolarized data, so σ_x should be removed

Reviewers' comments:

Reviewer #1 (Remarks to the Author):

"I appreciate the efforts that the authors invested into collecting new data, revising the manuscript and explaining their conclusions in the reply. I am now more convinced that the pressure-induced effect is real. I also consider that most of my concerns raised in the first report have been resolved. The only remaining point, which is also the one raised by the third referee, is the way the authors present "guides to the eyes" to their data that are sometimes biased and overemphasize the reported effect. These issues still remain in the new manuscript and should be resolved before the paper is published."

We appreciate these comments from the referee.

"Namely, in Fig. 2(g) the blue line is drawn below all blue (Mc) data points between 2 and 5 meV to enhance the impression that the blue and green lines coincide. Similarly, in Fig. 3(h) the peak in the blue line appears to be much sharper than the data suggest. In the middle (near T_N), it passes through the top points, completely ignoring the central point that lies much lower. On the left-hand side, however (~ 135 K), it is drawn much lower than the average, ignoring the data point at 135 K. These "tricks" in data presentation are bad practice. If there is no obvious fitting function that one can use to fit the data, one must ensure that the "guides to the eyes" that are drawn by hand pass through the middle of the dataset, reflecting the local "average" of the data points, otherwise the line is only misleading the eyes. The authors must fix their "guides to the eyes" accordingly in these two figures."

We appreciate very much these instructive comments and agree with the referee that we don't want to mislead the future readers. In the revised draft, we modified the guides to the eye so they are the true representation of the data. Thank you for keeping us completely honest on this detail. We appreciate very much for your careful reading of the paper.

"Apart from this remark, I consider that the revised version of the paper can be accepted for publication."

Thank you very much for your support.

Reviewer #2 (Remarks to the Author):

"The authors have addressed my questions. The paper is now suitable for publication."

Thank you very much for your support.

Reviewer #3 (Remarks to the Author):

"With this revised manuscript, the authors provided efforts, being more transparent, adding details and

descriptions, they even performed new measurements, their work is clearly appreciated. However, I have to say I am still not convinced by figure 3h, although I agree that in this version it is more objectively presented. The new figures, and the text improved accordingly, are helping. The general message looks more comprehensive.”

We appreciate these comments very much.

“I still have questions concerning the new figure 4 though, for which I had sometimes troubles to follow the description.

Are the $\theta/2\theta$ scans performed under pressure?”

Yes. This experiment was carried out using fixed pressure device as shown in supplementary Fig. 1. In the revised figure caption, we added a sentence to make this absolutely clear. We also added $P=20$ Pa in the figure to clarify the situation.

“Looking at figure 4d, I was wondering why σ_y was not appearing. There is no mention of it in the main text but I had the answer in the SI. It has not been measured, why? I believe it would have added proof, and it is always worth checking if it is consistent with what is seen for x and z.”

At the time when we did these measurements on BT-7, the neutron polarization analysis set up using He3 polarizer cannot do σ_y , and therefore these data were not taken. We added a sentence in the figure caption to make this clear. Nevertheless, we believe the conclusion of the results will be unchanged. This is confirmed by our unpolarized neutron diffraction Q-scan results and temperature dependent results shown in Fig. 4(e).

“Moreover, I found figure 4f not very clear, a more detailed description would be necessary.”

Thank you for making this comment. This is taken care of in the revised Fig. 4 caption.

“Finally, here are some minor comments:

- figure 1h should be 1g, and just below in the title of the caption, Ba2As2 should be BaFe2As2
- figure 4e is supposed to show unpolarized data, so σ_x should be removed”

Thank you so much for reading the paper so carefully. These typos are taken care of in the revised Figures.

Reviewers' Comments:

Reviewer #3:

Remarks to the Author:

Thank you for the corrections/improvements. I have no further comments/questions. I believe the present version of the manuscript is ready for publication.